# Prototyping an HEALTH DCAT-AP data catalogue to support population health indicator identification and quality assessment[*]

Rob Brennan[1,†], Junli Liang[1] and Akila Wickramasekara[1]

[1] ADAPT Centre, University College Dublin, Dublin 4, Ireland

## Abstract

This paper describes prototyping experiences in a population health use case of the draft HEALTH DCAT-AP specification for health data catalogues under the European Health Dataspaces Regulation. This included the development of a data catalogue metadata model, catalogue population via direct data entry and scraping of open data, and development of health indicator quality and feasibility reports. It was found necessary to extend the catalogue with new classes and properties for this use case, some of which were from the Data Privacy Vocabulary (DPV), and a number of limitations in the current HEALTH DCAT-AP specification draft were discovered. Stakeholders were generally positive in their assessment of the contribution of this novel structured approach to health data indicator discovery and assessment. This shows the potential for the semantic data governance infrastructure specified by the European Health Dataspaces Regulation to influence future data-driven decision making at all levels of European health services. The catalogue metadata model, report queries and data scraping code are all made available as open source resources for reuse by others. One new property has been added to DPV as a result of this work and it will feed into the HEALTH DCAT-AP standardisation process in the ETSI/TC Data. This paper describes a population health use case based on defining a health and wellbeing profile for older adults, data catalogue competency questions for this use case, a metadata model for the catalogue that meets these requirements, and a data quality feasibility and assessment reporting workflow along with stakeholder feedback.

## Keywords

W3C DCAT, EHDS, metadata, data quality, dataspace

## 1. Introduction

Many countries still struggle with health data management (Cresswell 2025) but the benefits of data-driven health planning are well known (Venkataramani 2020, Han 2020). The dominant method of strategic allocation of resources for population health remain on easy-to-interpret indicators or metrics manually created and validated by clinical experts (NAS 2020). There are a multitude of potential sources for these population level indicators, from national statistical agencies, charities, research institutes, hospital records or international agencies such as EuroStat. Typically when a planning a new health programme, a set of relevant and viable indicators must be assembled and subjected to peer review. This is a labour and knowledge intensive process that includes both data quality and clinical decision-making (NAS 2020).

Hence there is a need for a structured, repeatable approach to health indicator and dataset search and data quality appraisal, for example to feed into a wider indicator prioritisation process (McGlacken 2025). The introduction of the European Health Data Space (EHDS) Regulation (EHDS 2025) provides the necessary basis for standardised metadata across national and international health datasets such as health indicator sources. Standardisation in support of the EHDS is ongoing and Health DCAT-AP, an extension of the W3C DCAT (Data Catalogue) specification (Health DCAT-AP 23), has been proposed by the EHDS2 pilot and further developed by TEHDAS2 (TEHDAS2 2025). Thus the EHDS will stimulate the growth of national semantic data catalogues for health data and could enable new standardised governance methods and tools for secondary data use to support policy or planning applications such as population health. To date most of the focus of Health DCAT-AP has been on primary use i.e. patient care, see for example Gyrard et al.

---

[*] NXDG 2025: NeXt-generation Data Governance workshop co-located with 21st SEMANTiCS, 3-5 September 2025, Vienna, Austria

in cancer use cases (Gyrard 2025). Widespread use and testing of Health DCAT-AP remains to be seen in secondary use cases.

The research question studied in this paper is "*To what extent can a EHDS-compliant Health DCAT-AP data catalogue support population health indicator identification and quality assessment?*". The technical approach is to first develop a RDF-based model of health indicators and their source datasets using Health DCAT-AP as a basis. Then open data sources were scraped to populate the indicator and dataset catalogue. Finally a prototype semantic web toolchain was developed to query the catalogues to generate a quality assessment and feasibility report as input to the expert-based health indicator prioritisation and selection process. This work was carried out in collaboration with the Irish National Clinical Programme for Older People, the National Health Service Improvement Department and the National Health Intelligence Unit in the Irish Health Service Executive[1] (the body responsible for delivering health services nationally). This highlighted a number of limitations in the current Health DCAT-AP draft standard for this use case and some potential extensions.

The contributions of this paper are: i) the first documented development of a large scale secondary use application for Health DCAT-AP; ii) a set of reusable competency questions for population health indicator quality and feasibility assessment; iii) iv) a set of lessons learned from a large scale application of Health DCAT-AP in a National Health Service; and v) a set of open source scripts, RML mappings and SPARQL queries for our reporting toolchain.

The rest of this paper is structured as follows: §2 describes our use case, §3 gives an overview of related work, §4 describes our data catalogue model for population health indicators, §5 describes our case study-based evaluation and §6 provides brief conclusions.

## 2. Use Case

This work was carried out in the context of the need to develop a national Older Adult Health and Wellbeing Profile for Ireland to enable population-based planning at national, regional and local areas called integrated health areas, (IHAs). The profile would consist of a set of health indicators (metrics), typically with a name, definition, numerator and denominator, measurement unit, data source, timeframe, rationale and limitation. Each numerator and denominator could have seperate data sources and there are a large number of candidate datasets in Ireland from the Central Statistics Office, The Irish Longitudinal Study on Ageing (TILDA) a large-scale, nationally representative, longitudinal study on ageing in Ireland, the HSE National Health Intelligence Unit Core Indicator List, the Irish Hospital In-Patient Enquiry (HIPE) system, charities, and international sources such as the OECD or EuroStat. A seven step health indicator prioritisation process was developed to enable review and input from experts, patients, data publishers, international best practice, planners and policy-makers (McGlacken 2025).

However the wide variety of data sources and lack of established data catalogues or unified data governance processes meant that information was siloed on questions of data source quality and feasibility, for example:
- Does this dataset cover the appropriate population? (i.e. people who are aged 65+)
- Does this dataset support appropriate spatial and temporal granularity for this use case?
- Is this data updated frequently enough to fit the indicator?
- Is this data accurate and complete enough for population-based planning?
- How easy will it be to find and use this data?
- Are there data protection concerns for using this dataset?

If it was available, then a well maintained data catalogue could answer many of these questions which are orthogonal to the issue of the clinical suitability of a given indicator.

---

[1] https://about.hse.ie/

## 3. Related Work

A data catalogue is a repository and metadata management tool that provides an organised and searchable inventory of an organisation's data assets. It is a fundamental enabler of data governance in or between organisations. Data catalogues empower users to discover, understand, and leverage data for analytical purposes, reporting, and informed decision-making (Subramaniam 23). One of the leading technical specifications for data catalogues is the W3C Data Catalog Vocabulary (DCAT) that provides fundamental classes and properties for describing an organisation's data infrastructure in terms of datasets, dataset distributions, data services and data catalogues (DCAT 24). Since DCAT is an intentionally loose specification (to enable interoperability with minimal constraints), the SEMIC action within Interoperable Europe has developed an "application profile (AP)" for DCAT that includes additional constraints, e.g. cardinality, on the use of DCAT in EC data to ease interoperability (DCAT AP).

A key development for all health data sharing is the European Health Data Space (EHDS) Regulation[2] which came into force in March 2025. This will result in primary legislation supporting health data sharing for primary and secondary uses by 2030. However the EHDS outcomes are mainly legal, ethical and regulatory. Deployment relies on technical details based on the recommendations of the Data Spaces Support Center (DSSC)[3] which coordinates between many Data Space initiatives and projects such as Gaia-X[4]. Use of linked data knowledge graphs to organise machine readable data is central to the DSSC plans, as are DCAT (Data Catalog Vocabulary)-based data catalogues in dataspaces protocol specification of the International Data Spaces Association (IDSA)[5] referenced by the DSSC. HealthDCAT-AP (Application Profile) is being developed by the EU Health Data Spaces Pilot project as use of DCAT-AP is recommended by them for the EHDS. Given the sensitivity of health data there is a crucial role for security in the EHDS and there is emerging work on how this may be applied to sharing machine-readable knowledge models Julio 2023).

Since data protection concerns are central to sensitive heath data processing, part of the extensions provided by Health DCAT-AP are additional fields to cover this. These additional fields are taken from the Data Privacy Vocabulary (DPV) (DPV 2024). DPV is is designed to enable creation of machine-readable metadata about the use and processing of data, with an emphasis on personal data and associated legal requirements such as the GDPR, Data Governance Act and AI Act (DPV 2024).

## 4. Data Catalogue for Population Health Indicators

By holding a series of stakeholder workshops from January to May 2025 and examining the literature and public metadata for existing Older Adult Health Profiles such as NHS Fingertips[6] a set of requirements were developed for the data catalogue based on the public Health DCAT-AP draft. This resulted in a set of competency questions shown in Table 1. Four data quality and feasibility question areas were identified as most likely to be tractable for data catalogue-based assessment: completeness, precision, timeliness and data protection. Data protection is an issue that goes beyond typical data quality models but is very important to understand for data feasibility in projects like this.

A data catalogue metadata model (Fig. 1) was then developed with the guiding principles of: i) using the Health DCAT-AP sub-profile for sensitive data as a foundation; and ii) including minimal additional fields to answer the competency questions and stakeholder concerns. This resulted in 40 data fields being used for datasets. A set of application profile recommendations were also defined as per Health DCAT-AP with each field being marked mandatory, recommended or optional.

---

[2] https://health.ec.europa.eu/ehealth-digital-health-and-care/european-health-data-space_en
[3] https://dssc.eu/
[4] https://gaia-x.eu
[5] https://github.com/International-Data-Spaces-Association/ids-specification/releases/tag/2024-1
[6] https://fingertips.phe.org.uk/

**Table 1**

Competency Questions for a Population Health Data Profile

| Number | Domain/Question |
|--------|-----------------|
| 1 | Completeness |
| 1.1 | Are all mandatory and recommended fields present? |
| 1.2 | Are there datasets specified for the numerator(s) and denominator(s)? |
| 1.3 | For each indicator with numerator and denominator data sources, do the numerator and denominator datasets have data for i) all of Ireland and ii) for a target Healthcare Area? |
| 1.4 | For each indicator with data sources, do the numerator and denominator datasets have data for people aged 65 and over? |
| 2 | Precision |
| 2.1 | For each indicator with data sources, do the numerator and denominator datasets have data at the minimum temporal resolution required by the indicator? |
| 2.2 | For each indicator with data sources, do the numerator and denominator datasets have data at the geospatial resolution of national, health region, integrated heath area scales? |
| 3 | Timeliness |
| 3.1 | For each indicator with data sources, do the numerator and denominator datasets have data published at the frequency required by the indicator calculation? |
| 3.2 | For each indicator with data sources, do the numerator and denominator datasets have data published at the frequency required by the reporting style? |
| 4 | Data protection |
| 4.1 | Does each dataset used by indicators contain personal data, sensitive personal data or pseudonymised personal data? |
| 4.2 | Does any dataset potentially contain personal data? |
| 4.3 | Is there any dataset with personal data that or potential personal data that is not controlled by the HSE? (i.e. needs data sharing agreement) |

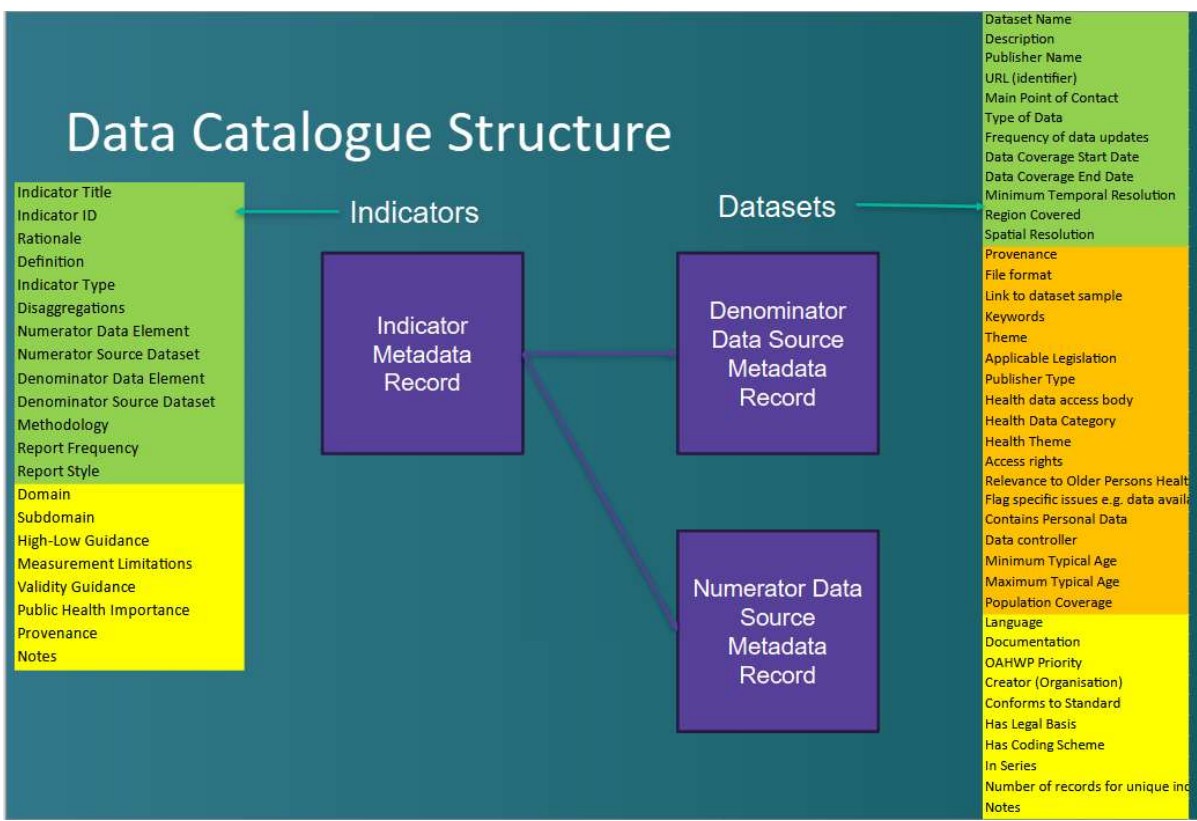

**Figure 1:** Data Catalogue Structure Showing Indicator Record, Dataset Record (green fields are mandatory, orange recommended and yellow optional).

**Table 2**

New Metadata Fields Added to Health DCAT-AP for this use case

| Field | Description | Source |
|---|---|---|
| Status | Describes the record's status within the indicator prioritisation process. Values: Include in Catalogue, Exclude from Catalogue, Under Consideration, Exclude from Profile, Include in Profile | New |
| Contains Personal Data | Indicates association with Personal Data [in this dataset] | dpv:hasPersonalData |
| Data Controller | Indicates association with Data Controller [for this dataset under GDPR] | dpv:hasData controller |

## 5. Evaluation Case Study

The data catalogue model described above was tested by deploying in support of the use case described in section 2. The goal was to provide a structured way to record information about the large number (over 1000) datasets and indicators being considered in the process of defining the final set of indicators for the Older Adult Health and Wellbeing Profile. This was a seven step process (see McGlacken et al. ) and for three of the steps there was a need to have a data quality and feasibility report generated from the metadata.

To populate the catalogue a workflow (Fig. 2) was defined using web scraping scripts to CSV files, some manual data entry and semi-automated valuation before uplift using R2RML. The

RDF-based data catalogue had SPARQL scripts created to answer the competency questions in Table 1.

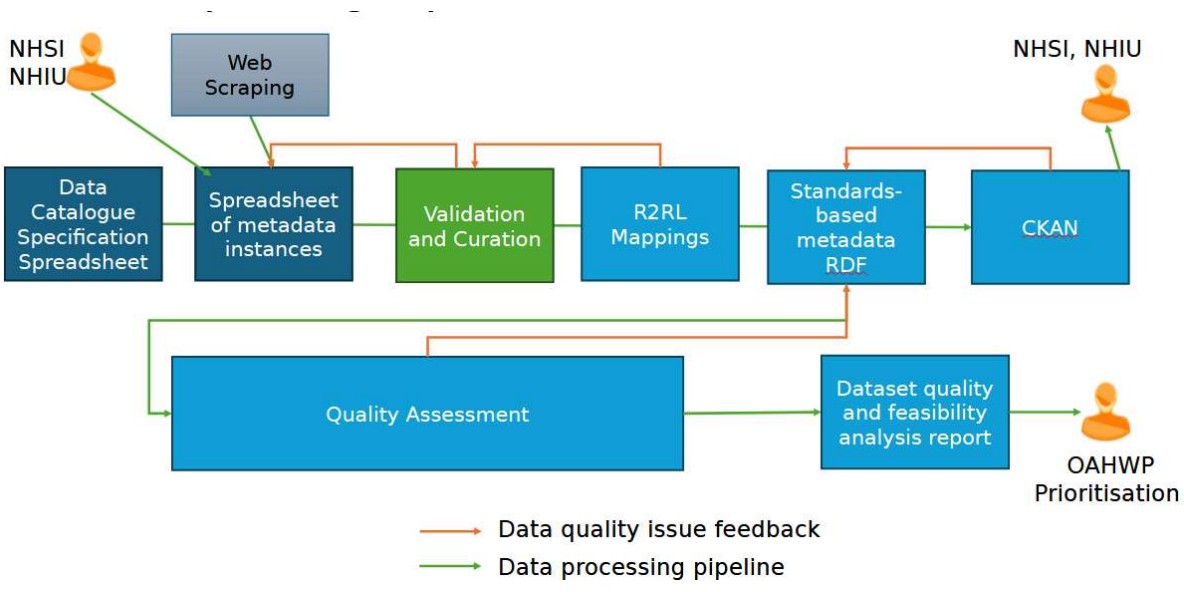

**Figure 2:** Metadata Ingestion, Validation, Assessment and Reporting Pipeline

## 5.1. Evaluation and Self Reflection

There were a set of HEALTH DCAT-AP Issues identified:

1. Insufficient attention is provided to data protection as can be seen from our additional fields. In some cases the reuse of DPV properties by HEALTH DCAT-AP assumes a string as the range of the property when a range class is defined in DPV and these should be made compliant.

2. Spatial resolution of datasets is identified in metres. It would be much more useful for population health to be able to specify national, county, NUTS2 regions and also new regional areas.

3. Spatial coverage is limited to regions that are modelled in Geonames. This does not include regional subdivisions like the IHAs defined by the Irish health service. It is not a sustainable solution to have coverage definitions managed by a 3rd party private organisation so other spatial spatial region definition authorities like National Mapping Agencies should be allowed by the specification.

4. In many cases Health DCAT-AP defers to WikiData for the definition of categorical field codes. This is a fine pragmatic solution but it should not be the only source allowed e.g. National authorities should be able to publish their own IDs.

6. No Publisher codes were defined by Health DCAT-AP. A suggessted set of codes we found by searching the EHDS text was: National Public Health Institute, National Mapping Agency, Statistical Agency, Hospitals and Healthcare Providers, Universities and Research Centers. Health Departments. Community-based and Clinical Care Organisations.

## 6. Conclusions

This work has shown that the EHDS gives a large opportunity to improve the data governance infrastructure for all healthcare data governance. In particular the Health DCAT-AP draft specification gives a strong basis for implementation but will need to be refined further for

deployment, especially for secondary use cases. The results of this will will be fed into the standardisation process both nationally and at ETSI.

## Acknowledgements

This research was conducted with the financial support of the Health Service Executive National Clinical Programme for Older People Research Award 2024 (NCPOP RA3/2024) and the Research Ireland ADAPT Research Centre in the RI Research Centres Programme (Grant 13/RC/2106_P2), For the purpose of Open Access, the authors have applied a CC-BY public copyright license to any author accepted manuscript version arising from this submission.

We would like to extend our gratitude and appreciation to the stakeholders for their feedback and engagement with this work. Special thanks to our colleagues in the HSE for their invaluable contributions to this paper: Thereese McGlacken, Stephen Barrett, Jacinta Mulroe, Teresa Bennett, Declan McKeown, Gerardine Sayers, Mary Browne, Aparna Keegan, Graham Hughes.

## Declaration on Generative AI

The authors have not employed any Generative AI tools.

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

# A. Online Resources

The HSE project scripts, mappings and RML can be downloaded from this repository: https://github.com/junli-liang-johnny/hse-scripts