# OpenReview forum: "Prototyping an HEALTH DCAT-AP data catalogue to support population health indicator identification and quality assessment"
_SEMANTiCS.cc/2025/Workshop/NXDG — NXDG 2025_

### Official Review · ~Chang_Sun1 · 2025-07-16
**Nice paper addressing an timly and important topic**

**Rating:** 6
**Confidence:** 3

**Review:**

This manuscript is very well written with a nice structure make it easy for readers to follow. It addresses an interesting and important research question "To what extent can a EHDS-compliant Health DCAT-AP data catalogue support population health indicator identi􏰃cation and quality assessment". I think this is well-suited for the workshop audience.

However, the methods part could be more elaborative. It is unclear to me how the data catalogue structure was designed and validated? The elements in the indicator metadata record need some explanation. Among these elements, some of them are mandatory and some of them are recommended and optional, how are they classified to these different requirements? Are these elements for health cases in general or only for this use case? The results session is brief and only observations and lessons learnt are reported in the paper. It would be great if some suggestions / potential solutions can be proposed with the self-reflections.

---

### Official Review · ~Kimberly_Garcia1 · 2025-07-22
**This paper presents the results of utilizing Health DCAT-AP in a large prototype of health indicator in Ireland; after implementing a use case, and trying to answer a set of competency questions, a list of limitations is presented. The objective is to enhance Health DCAT-AP through these findings.**

**Rating:** 8
**Confidence:** 4

**Review:**

This contribution is very interesting. However, more context could improve some of the sections to make the paper self-contained

My recommendations per session:

The abstract would benefit from a higher-level explanation introducing why we should care about this work (i.e., data-driven health planning), before going directly to all the results and contributions of the paper.

Perhaps adding the research question mentioned in the introduction will help to clarify why it is important to work on this.

Section 2: To make the paper more informative, it would be great to get more context of at least one of the health indicators that is being tackled, or at least a reference. This would also make clearer and stronger the motivation on why we care about this problem. Moreover, getting more information about a few (or at least one) health indicator, could help reader understand how the extensions you propose in Section 4 are generalizable to other use cases.

Section 4: Could you provide more information about the workshops? Who are the stakeholders? What methodology was used to obtain the set of requirements? How many stakeholders were involved? how did you chose the competency questions to address?

Section 5.1: Evaluation and self-reflection could be more contextualized, now it reads as a set of limitations. However, there is little contextualization to understand these limitations in depth. The readability of the section could be improved by enhancing the self-reflection portion: why is it important to address these limitations? What do they hinder when tackling the use case in section 2?

Small issues:

Page 3 typo in last paragraph before sec 3.

Page 5 typo “the large number (over 1000) datasets” of is missing.

Figure 2 is a screenshot, the quality of the image could be improved, it even shows an error on the legend “CKAN”.

Formatting of the 5.1 enumeration.

---

### Decision · Program_Chairs · 2025-07-25

Accept